# Assessment of Exercise Capacity in Post-COVID-19 Patients: How Is the Appropriate Test Chosen?

**DOI:** 10.3390/life13030621

**Published:** 2023-02-23

**Authors:** Rodrigo Torres-Castro, Rodrigo Núñez-Cortés, Santiago Larrateguy, Xavier Alsina-Restoy, Joan Albert Barberà, Elena Gimeno-Santos, Agustin Roberto García, Oriol Sibila, Isabel Blanco

**Affiliations:** 1Department of Pulmonary Medicine, Hospital Clínic—Institut d’Investigacions Biomèdiques August Pi i Sunyer (IDIBAPS), University of Barcelona, 08036 Barcelona, Spain; 2Department of Physical Therapy, Faculty of Medicine, University of Chile, Santiago 8380453, Chile; 3Physiotherapy in Motion Multispeciality Research Group (PTinMOTION), Department of Physiotherapy, University of Valencia, 46003 Valencia, Spain; 4Servicio de Kinesiología y Fisioterapia, Hospital de la Baxada “Dra. Teresa Ratto”, Paraná 3100, Argentina; 5Facultad de Ciencias de la Salud, Universidad Adventista del Plata, Libertador San Martin 3103, Argentina; 6Centro de Investigación Biomédica en Red de Enfermedades Respiratorias (CIBERES), 30627 Madrid, Spain; 7Barcelona Institute for Global Health (ISGlobal), 08036 Barcelona, Spain

**Keywords:** exercise capacity, tests, post-COVID-19

## Abstract

There is a wide range of sequelae affecting COVID-19 survivors, including impaired physical capacity. These sequelae can affect the quality of life and return to work of the active population. Therefore, one of the pillars of following-up is the evaluation of physical capacity, which can be assessed with field tests (such as the six-minute walk test, the one-minute standing test, the Chester step test, and the shuttle walking test) or laboratory tests (such as the cardiopulmonary exercise test). These tests can be performed in different contexts and have amply demonstrated their usefulness in the assessment of physical capacity both in post-COVID-19 patients and in other chronic respiratory, metabolic, cardiologic, or neurologic diseases. However, when traditional tests cannot be performed, physical function can be a good substitute, especially for assessing the effects of an intervention. For example, the Short Physical Performance Battery assessment and the Timed Up and Go assessment are widely accepted in older adults. Thus, the test should be chosen according to the characteristics of each subject.

## 1. Introduction

The coronavirus disease (COVID-19) pandemic has challenged health systems worldwide, affecting more than 660 million people, with more than 6.7 million deaths as of January 2023, according to the World Health Organisation [1]. Although most people infected by the SARS-CoV-2 virus develop asymptomatic or mild disease, about 20% develop severe disease requiring hospitalisation [2]. Acute respiratory distress syndrome is one of the main complications of COVID-19 and some cases (e.g., cases with mechanical ventilation or septic shock requiring vasopressors) require admission to the intensive care unit (ICU) [3]. In this context, prolonged hospital stays can have negative effects on the physical health of patients due to bed rest and poor mobility [4]. In fact, a previous study in patients hospitalised for COVID-19 described a marked loss of muscle strength after hospital discharge, even in cases without previous locomotor impairment [5].

Although COVID-19 is primarily a respiratory condition, it affects other systems, such as the cardiovascular, musculoskeletal, and neurological systems, thus generating a broad range of sequelae that affect COVID-19 survivors in the short, medium, and long-term [6,7,8]. Among the most commonly reported sequelae are symptoms such as fatigue, dyspnoea, and, in some cases, even a marked deterioration in physical capacity [9,10]. These sequelae affect the quality of life and return to work in the active population [11]. Thus, it is estimated that the long-term sequelae of COVID-19 in patients after hospital discharge can reach 76% of cases at six months (i.e., at least one symptom at follow-up, such as fatigue, muscle weakness, sleeping difficulties, anxiety, depression, or others), with higher prevalence in women [12]. The considerable proportion of cases with sequelae following acute infection is of concern to both patients and rehabilitation providers, as many patients are likely to require long-term support and treatment [13].

Due to the significant number of patients with sequelae [14,15,16], the different health systems and scientific societies have generated follow-up programmes that focus primarily on symptoms, lung function, imaging, and physical capacity [17,18,19]. One of the pillars of following-up is the evaluation of physical capacity, which can be assessed with field tests (such as the six-minute walk test (6MWT) or the 1-min sit-to-stand test (1 min-STST), among others) or laboratory tests (such as cardiopulmonary exercise test (CPET)) [4,20,21,22,23]. The great advantage of these tests is the analysis of the interactions between the cardiovascular, respiratory, and musculoskeletal systems [20].

These tests can be performed in different contexts and have widely demonstrated their usefulness in evaluating physical capacity in other chronic respiratory, metabolic, cardiological, or neurological diseases [20]. However, to provide the best information, the test must be chosen according to the characteristics of each subject, the objective of the test, the setting, and the physiological expected response [24]. Our objective was to review the current literature to describe the different assessments that can be performed on these patients and thus decide which is the most appropriate option according to the setting, the objective, and the expected response.

## 2. Laboratory Test in Post-COVID-19 Patients

### Cardiopulmonary Exercise Test

The CPET is the gold standard for assessing exercise tolerance: it provides additional information about cardiopulmonary function and supports the prescription of aerobic exercise [25]. CPET provides objective and subjective information from different systems (ventilatory, gas exchange, circulatory, and muscle/metabolic variables) to symptom scores [26]. Through the joint analysis of exhaled gases, the work and/or effort performed, and the behaviour of the hemodynamic variables, a more complete functional assessment can be obtained [25]. However, analysis and interpretation require the combination of CPET results with other clinical and laboratory findings [27,28]. On the other hand, verbal information from the patient, including the reason for stopping exercise and other symptoms that may arise during CPET, can be helpful in evaluating exercise intolerance and the underlying mechanisms of exertional dyspnoea [27,28].

Aerobic capacity, measured by peak oxygen uptake (VO_2_ peak) and ventilatory efficiency, quantified commonly by the minute ventilation/carbon dioxide production (VE/VCO_2_) slope, are two of the most-established measures obtained from the CPET [29]. In COVID-19, the cardiac and skeletal muscle systems can be negatively impacted; for this reason, it is not surprising to see a diminished VO_2_ peak in individuals infected, particularly those who were hospitalised due to the increased pathophysiologic severity of the viral infection [29].

There are numerous protocols that can be used for both healthy individuals and those with diverse chronic diseases. These protocols are used to prescribe steady-state aerobic exercise (commonly walking or running) or interval exercise [25]. It is best that CPET is performed with a ramp protocol, with a duration between eight and twelve minutes, and on the same device that the patient uses for training (namely a cycle ergometer for cyclers and a treadmill for walkers/runners) [25].

## 3. Field Test in Post-COVID-19 Patients (Figure 1)

### 3.1. Six-Minute Walk Test

The 6MWT is a simple, easy, and inexpensive test. It is well standardised, has reference equations for interpretation, and is widely used in clinical practice [20]. It provides relevant information on submaximal (in some cases, maximal) exercise capacity and is useful for monitoring patients with cardiovascular and respiratory diseases [20]. The main outcome is the maximal distance that a patient may cover walking for six minutes (6MWD) [20]. On the other hand, the execution of the 6MWT requires a corridor of 20-to-30-m, which is often unavailable in hospitals or rehabilitation centres [30] and even less at home.

**Figure 1 life-13-00621-f001:**
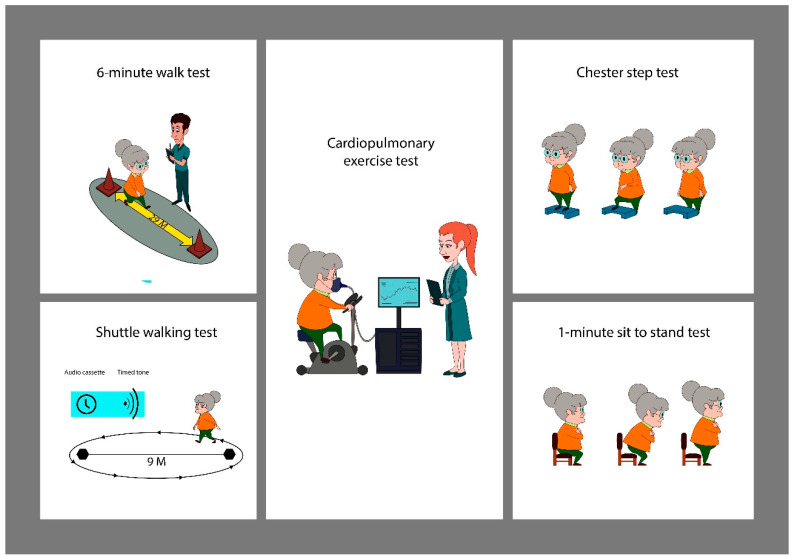
Schematic representation of different laboratory and field tests.

During the test, standardised instructions and directions are usually given, which include: walk down the aisle between the markers, as many times as you can in six minutes; report each time one minute passes and at six minutes ask the patient to stop where they are; the patient may slow down, stop, and rest if necessary, but should resume walking as soon as they can; and the patient should not run or jog. Finally, the evaluator should resolve any doubts of the patient before starting the test [20].

The American Thoracic Society and European Respiratory Society (ATS/ERS) recommendation [20] supports that the 6MWD is very sensitive to methodology variations, including track layout and length changes. However, modifications and space limitations are the main reason for using a shorter-than-recommended walkway [31]. A recent study compared the 10-m and 30-m tracks, finding that patients with chronic non-communicable diseases walk about 70 m less on the 10-m track, which also exceeds the minimally clinically significant difference [32]. This effect may be exacerbated in patients with post-COVID-19 with prolonged bed rest and older age, which may affect balance and gait patterns and/or strategies [33]. In fact, a recent investigation identified that 70% of post-COVID-19 patients had values below the estimated 6MWD one year after hospital discharge [34]. In addition, patients with a higher comorbidity burden had worse 6MWT performance. Specifically, a higher Charlson comorbidity index score one month after hospital discharge was an indicator of a shorter distance walked on the 6MWT at 1-year follow-up evaluations [34]. On the other hand, González et al., in a sample of patients with COVID-19 who required admission to the ICU, found a significant reduction of 128 m in the distance walked in the 6MWT compared with reference values at 3 months after hospital discharge [35]. In addition, patients with more severe alterations in thoracic computed tomography showed worse pulmonary function and presented more degrees of desaturation in the 6MWT, indicating poor exercise tolerance [35]. Therefore, 6MWT may be a good tool to identify cases with greater functional impairment after hospital discharge, and performance may be worse in those cases with pulmonary structural abnormalities. Finally, 6WMT may also be a good outcome to evaluate the effects of rehabilitation programmes for post-COVID-19 patients with the aim of improving measured exercise capacity. For example, in a recent meta-analysis, Chen et al. found that the pooled estimate of the effect of pulmonary rehabilitation for post-COIVD-19 patients at 6MWT was 50 m, that is to say, higher than the minimum clinically important difference recommended in chronic lung disease [36].

### 3.2. Sit-to-Stand Test

The sit-to-stand test (STST) is a quick and easy-to-use field test in which participants are instructed to stand and sit up from a chair with their arms crossed over their chest as many times as possible for a set amount of time (i.e., 30 s or 1 min) [21,37]. The STST is a surrogate test used in patients who cannot walk in the standardised conditions described for the 6MWT (e.g., in a 30 m corridor) [38].

There are at least three different versions of the STST: (1) the five times STST (5-STST), which measures the time it takes a person to stand up and sit down five times; (2) the 30-s STST (30 s-STST), which measures the number of times a person stands up and sits down in 30 s; and (3) the 1 min-STST, which measures the number of times a person stands up and sits down in 60 s [38]. The one-minute version is particularly effective since, in other pathologies, there is a good correlation with the 6MWT (e.g., chronic obstructive pulmonary disease (COPD)) [39]. Although the technical gesture of the 5-STST or the 30 s-STST is the same because of the test expended time, the 1 min-STST significantly stresses the lower extremities but does not use the cardiorespiratory reserves in the same way [39]. Therefore, these shorter exercise tests are commonly used to assess the strength of the lower extremities or predict falls in older adults [40,41]. For example, the 30 s-STST was proposed as a valid procedure to assess lower limb muscle power in older adults, which would be more strongly related to physical performance (e.g., maximal walking speed) than STST performance based solely on the number of repetitions [42]. Furthermore, 30 s-STS power is positively associated with pectoral muscle thickness on chest CT in both male and female COVID-19 survivors [43]. Thus, this test may indicate global muscle wasting and is a good alternative compared to more sophisticated instruments, as it requires less time, equipment, and space. The 30 s-STST can also be used to plan the early stages of rehabilitation in survivors of COVID-19. Moreover, the 30 s-STST has proven to be a feasible and safe test to apply in the telehealth setting and is related to the prolonged sequelae of COVID-19 (e.g., fatigue, dyspnoea, and pain) in non-hospitalised cases. Therefore, health teams can use this modality when the assessment of the physical sequelae of COVID-19 in a face-to-face setting is not possible (e.g., due to geographical and socio-economic barriers) [44].

The 1 min-STST is a good tool for assessing physical capacity and exercise desaturation in post-COVID-19 patients [4] and, unlike other tests, it has the advantage that it needs to be performed only once because it has a marginal learning effect [45]. A sturdy chair of standard height (e.g., 46 cm) without armrests, placed against a wall, is recommended for the test. It is recommended that the test be conducted with only the investigator and patient present to avoid distractions. During the test, participants should be verbally encouraged to complete as many standing cycles as possible in 60 s at their own pace. Participants are not allowed to use their hands or arms to push the chair seat or their body. For data analysis, it is recommended to use the healthy adult population-based reference values previously reported by Strassmann et al. [46]. In post-COVID-19 patients it has been reported that up to 83% of patients can complete this test and 90% were below the 25th percentile, relative to baseline values [4]. In addition, 32% of patients have a decrease in pulse oxygen saturation equal to or greater than four points. Thus, the 1 min-STST was able to discriminate between those with and without prolonged hospital stay (i.e., hospital stay of >10 days) in terms of exertional desaturation [4].

Given the moderate correlation between the 1 min-STS and the 6MWT, it is recommended for rehabilitation and telerehabilitation programmes [23,38,47]. This is an important point in the COVID-19 pandemic as many services have had to initiate remote monitoring and rehabilitation programmes in rehabilitation services [48].

### 3.3. Chester Step Test

The Chester Step Test (CST) is a standard submaximal incremental field test with the advantage of requiring little space and having highly portable equipment [49]. Therefore, it can also be performed in different settings, such as in an ICU, an outpatient clinic, or a home rehabilitation programme. The CST uses a standardised step from 15 up to 30 cm in height (for ascending and descending) and a progressive sequence of auditory stimuli to determine the speed at five levels (first stage: 15 steps/min; second stage: 20 steps/min; third stage: 25 steps/min; fourth stage: 30 steps). Each stage lasts two minutes (a total test of 10 min) [50]. The CST was initially designed to estimate exercise capacity in healthy subjects. In fact, the CST has proven to be a valid test for estimating aerobic capacity, with a high correlation with the (r = 0.92) peak oxygen uptake (VO_2_ peak) [51]. The CST has also been widely used in patients with chronic and acute respiratory diseases (e.g., COPD and severe acute respiratory syndrome) [39,52,53,54]. Furthermore, a recent investigation identified that the number of steps achieved in the CST correlates moderately and significantly with the distance walked during the 6MWT in post-COVID-19 patients, being a good alternative when 6MWT cannot be performed [55]. Moreover, the CST has proven to be a reproducible tool (ICC = 0.993) to assess exercise capacity and exertional desaturation in post-COVID-19 patients. Post-COVID-19 patients can achieve a median of 123–129 steps on the CST around six months post-infection. Between 24% and 30% of participants may have a clinically significant decrease in SpO_2_ (i.e., ≥4 points) [54]. Therefore, CST could also be a helpful test to guide rehabilitation teams in decision-making. For instance, for exercise prescription (resistance training) and aerobic capacity improvement, especially in those patients with COVID-19 with persistent symptoms [54].

### 3.4. Shuttle Walking Test

The incremental shuttle walking test (ISWT) is an externally paced maximal exercise test [20,56]. The ISWT was developed to simulate a CPET using a field walk test [57]. The patient must walk on a 10-m track while a series of beep sounds indicate the walking speed, which will increase every minute [20,57]. The patients walk as long as they can until the symptoms limit the exercise or they can no longer keep up with the walking speed, at which point the test is terminated. The distance walked and the speed achieved are recorded [20].

The endurance shuttle walking test (ESWT) can be performed based on the maximum level reached in the ISWT, which consists of walking at a fixed speed (usually 70–85% of that reached in the ISWT) for the maximum time possible [20,57]. The ISWT shows a linear change of lung gas exchange, including the VO_2_ peak [58].In addition, the distance walked in this test has been reported to be reliable and a good indicator for predicting re-hospitalisations in patients with COPD [59].

The ISWT has 12 levels, and each level takes one minute. It is necessary to use a pre-recorded audio file with sound signals. The recording consists of a signal for the start of the evaluation, a signal for the start of the level, and periodic signals on each shuttle. Three consecutive signals indicate the beginning of each level, and the patients try to reach the opposite cone before one periodic signal [60]. As the number of walking shuttles gradually increases with the level and the periodic cues shorten, patients walk faster and are encouraged to walk as many shuttles as possible. The trial is terminated if patients experience dyspnoea, are unable to maintain gait pace at the cue, or if the oxygen saturation level drops below 80% [60]. However, when the beep sounds, if the patient comes within 0.5 m of the cone, they are given the opportunity and encouragement to walk to the next shuttle [60].

So far, it has been used in a study in which a three-week pulmonary rehabilitation programme was conducted in 50 post-hospitalised COVID-19 patients, half of whom were in critical condition [61]. Critically ill patients improved their walking duration in almost three times assessed with ESWT (460 vs. 1200 s) [61].

### 3.5. Other Tests

In addition to the laboratory or field tests traditionally used in other diseases, other tests are being reported in post-COVID-19 patients, mainly tools developed for the older population, such as the Short Physical Performance Battery (SPPB) and the Timed Up and Go (TUG) [23,62].

The SPPB is a tool based on three timed tasks: (1) standing balance, (2) gait speed, and (3) chair standing test [63]. On the other hand, the TUG is a timed test that assesses an individual’s ability to get up from a chair, walk three meters (as fast as possible), turn around, and sit back down in the chair [64]. Both have the advantage of assessing key elements of functional capacity (i.e., balance and gait speed), making them widely accepted assessments in older adults. For this reason, it is essential, in addition to knowing the tests, to decide which is the best according to the target population and stage of disease progression. However, it is important to emphasise that these tests have been developed more than exercise tolerance to measure physical function. However, when it is impossible to perform traditional tests, physical function may be a good substitute, especially to assess the effects of an intervention.

### 3.6. Integration of Field Tests with Biological Signals

Several composite indices can integrate 6MWD and SpO_2_ changes [65,66,67,68]. The desaturation distance ratio (DDR) integrates the 6MWD with the desaturation area (DA) calculated from either a maximal theoretical value of 100% (original DDR) [65] or the actual SpO_2_ value of the patient measured at rest (new DDR) [68]. Alternatively, the distance saturation product (DSP) is the product of the 6MWD and the lowest SpO_2_ value determined during the test [69]. These composite indices predict mortality in chronic respiratory diseases [70,71,72,73] and can be useful in post-COVID-19 patients.

### 3.7. Factors That Affect the Exercise Capacity

Undoubtedly, there are several factors, either from the patient or from the place where the test is performed, that affect the patient’s physical capacity. The following factors must be considered for choosing the appropriate test:Age: The age of the patient is directly related to the functional capacity. The first published reports of physical capacity had a mean age of more than 70 years [74,75], particularly in the European population, showing low success rates when performing tests that required independent walking, such as the 6MWT. On the other hand, many of these patients may present frailty, so in this case, it is advisable to perform physical function tests such as the SPPB [8,62] (Table 1).Frailty: The literature has shown that COVID-19 infection was associated with functional decline in at least one-third of survivors [76]. On the other hand, recent evidence suggested that elderly COVID-19 patients have a high incidence of frailty, and frailty is detrimental to COVID-19 prognosis. In this scenario, the frailty assessment can be assessed, ideally with objective and multidimensional tools [77]. Physical and multidimensional frailty are both predictors of outcome and important moments to decide on rehabilitation programmes for this population, especially for the elderly [77].Balance: People who have been hospitalised for long periods develop muscle weakness, affecting balance and increasing the frequency of falls [78]. Balance is a predictor of a person’s ability to walk [79]. Additionally, if the patient is an older adult, it will be even more difficult to turn and accelerate in each lap in the 6MWT or SWT [80].Cognitive state: The cognitive state can be affected in post-ICU patients due to the use of medicaments and/or the presence of delirium [81]. Cognitive dysfunction produces a greater affectation of the activities of daily living and decreases the possibilities of optimal rehabilitation [82]. In addition, we must consider that all physical capacity assessment tests require the patient to follow instructions [20].Infrastructure: The current ATS/ERS recommendation supports that the 6MWD is very sensitive to methodology variations, including track layout and length changes [20]. Moreover, a recent study showed that the difference between the two protocols could reach almost 70 m apart [32]. The reasons to obtain different distances walked are multiple if the length of the corridor is not the same. Firstly, patients slow down the walking speed through the cone, decreasing the final distance. For example, a person who walks 600 m will only turn 20 times in the 30-m tracks but will turn 60 times in the 10-m corridor [24].

In summary, if it is possible to perform a CPET in an exercise testing laboratory, this is the recommended test to investigate the cause of persistent symptoms or to prescribe exercise training for a rehabilitation programme. However, if this scenario is not possible, the field tests such as the 6MWT or the 1-min-STS (depending on the infrastructure) are the best options for testing the functional capacity of patients without mobility or balance problems. On the other hand, if we are dealing with an older patient and/or with balance problems, the SPPB or the TUG are likely the more functional tests.

## 4. Practical Considerations

Once we decide which test is best, there are certain important clinical aspects to consider, regardless of which test is chosen. In this section we will try to answer the most frequently asked questions.

*When should I carry out the physical capacity evaluations?* The decision of the ideal moment to carry out these evaluations will depend on the objective. If the goal is the follow-up, it is usually conducted between 8 and 12 weeks after discharge [19,83]. However, if the objective is to evaluate the effect of rehabilitation, it is advisable to perform it before (baseline evaluation) and after the programme (post-intervention evaluation) [84]. If the programmes are short, as has happened in the first reports in the literature [85,86], it does not make sense to carry out interim evaluations.*How should I test if my patient uses supplemental oxygen?* The use of supplemental oxygen will depend on the extent of lung damage caused by the infection. There are several possible scenarios as to why a patient may need to undertake the test with supplemental oxygen. One of the critical points to consider is who carries the oxygen device. If the objective is to evaluate the impact of an intervention such as the use of a drug or the effect of a rehabilitation programme, it is better if the technician carries the device (both the compressed gas cylinder or portable oxygen concentrator) so that its weight does not influence the final result of the test [87]. On the other hand, if we want to evaluate physical performance through an activity of daily living, it is recommended that the patient carries the device, especially if the patient uses home supplemental oxygen such as portable oxygen concentrator, and in this way we can evaluate the impact of its transfer on our patient [88].*Does the patient have to wear a mask?* In the current pandemic context, we recommend following the local safety guidelines. If these do not exist, we recommend the use of a surgical mask. Although these masks slightly increase the sensation of dyspnoea, they do not affect the final result [89]. Notably, the use of FFP2 or N95 masks has been shown to decrease performance in field tests, as well as modifying exhaled gases [90]. This point is of crucial consideration in patients who have hypercapnia.*Should I continuously monitor the biological signals?* Yes. One of the main problems that post-COVID-19 patients have is desaturation on exertion, so the patient must be monitored during the entire test [4]. The lowest saturation should be registered if it is less than 90% and if the drop is greater than four points [91]. Another variable that can provide indirect information on cardiovascular health isheart rate recovery (HRR) per minute. HRR after one minute of less than 10–15 points is related to morbidity and mortality in various chronic diseases [92,93].*When should I stop the test?* Field and laboratory tests must include strict monitoring of biological signals. If the heart rate exceeds the established maximum value or if the oxygen saturation is ≤80%, for the test must be stopped [20]. On the other hand, we must be attentive to the behavior of the patient, particularly with respect to signs indicative of any possible complication such as excessive sweating, feeling dizzy, or similar [20].*What should I do if the patient has technical aids?* If the patient has a walking aid of any form, such as a cane or a crutch, they must use it as prescribed by the physician, and it is recommended to perform the test using the technical aid. These devices usually help maintain balance by expanding the base of support, so not using them puts patient safety at risk.

## 5. Conclusions

The evaluation of exercise tolerance is one of the pillars of following-up post-COVID-19 patients and can be assessed with field tests, such as the 6MWT, the STST, the CST, the SWT, or with laboratory tests (CPET). These tests can be performed in different contexts and have adequately demonstrated their usefulness in the assessment of physical capacity both in post-COVID-19 patients and in other chronic respiratory, metabolic, cardiological, or neurological diseases. Thus, the test should be chosen according to the characteristics of each subject.

## Figures and Tables

**Table 1 life-13-00621-t001:** Level of recommendation for each evaluation at discharge according to age group and comorbidities.

Assessment	<70 Years without Comorbidities	<70 Years with Comorbidities	>70 Years with Comorbidities	>70 Years without Comorbidities
CPET	☆☆☆	☆☆	☆	☆☆
6MWT	☆☆☆	☆☆☆	☆	☆☆
1 min-STST	☆☆☆	☆☆☆	☆	☆
30 s-STST	☆☆	☆☆	☆☆	☆☆
5-STST	☆	☆	☆☆☆	☆☆☆
CST	☆☆☆	☆☆☆	☆	☆
SPPB	☆	☆☆	☆☆☆	☆☆☆
TUG	☆	☆	☆☆	☆☆☆

Abbreviations; ☆☆☆: high recommendation: ☆☆: moderate recommendation; ☆: low recommendation; CPET: Cardiopulmonary exercise test. 6MWT: Six-minute walk test. 1 min-STST: 1-min Sit-to-stand test. 30 s-STST: 30 s Sit-to-stand test. 5-STST: Five times sit-to-stand test. CST: Chester step test. SPPB: Short physical performance battery. TUG: Timed Up and Go.

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
