# Peer review of "Assessment of Exercise Capacity in Post-COVID-19 Patients: How Is the Appropriate Test Chosen?"

_life, 2023, doi:10.3390/life13030621_

Round 1

Reviewer 1 Report

The Authors describe an important issue in post-covid evaluation. I found the manuscript of interest.

I suggest to describe and to discuss the concept of frailty that is another important moment in the evaluation of Covid patient. We know that physical and multidemensional frailty are both predictor of outcome and important moment to decide on rehabilitation programs for this population, especially for the elderly. 

Curcio F, De Vita A, Gerundo G, Puzone B, Flocco V, Cante T, Medio P, Cittadini A, Gentile I, Cacciatore F, Testa G, Liguori I, Abete P. Reliability of fr-AGILE tool to evaluate multidimensional frailty in hospital settings for older adults with COVID-19. Aging Clin Exp Res. 2022 Apr;34(4):939-944. doi:

Author Response

Dear reviewer

Thank you for your revision. Our response is in the attached document.

Best wishes

Rodrigo Torres Castro

Reviewer 2 Report

Dear authors,

I found this topic relevant and interesting for international audience especially during and after COVID 19 pandemic. However, your writing and your findings has to be presented in scientific way. Scope of this journal is to publish fundamental themes in life science, from basis to applied research. Your manuscript is very good prepared professional paper. If you want to prepare review paper you have to follow the rules for it and present how you extract research from the different database (you can use Prisma chart). Second major issue is 29.2% of overlapping with other literature, and this has to be reduced. At the end I give you full support to continue with this topic and prepare review article good enough to be published. 

Author Response

(The authors gave the same response as above.)

Reviewer 3 Report

Dear Author(s)

1. There are grammatical errors. Please re-edit the text.

2. Please pay attention to the full name of each abbreviation. For example, "cardiopulmonary exercise test" should be written as "CET" on page 5., etc.

3. In my opinion, this manuscript can be accepted as a mini-review, not a review. Therefore, you should reduce references and perform other formates for mini-review.

Author Response

(The authors gave the same response as above.)

Reviewer 4 Report

Nice work with practical conclusions. I don`t agree the self citing 7,9,10,11. I think you should use the original source in that articles.

Author Response

(The authors gave the same response as above.)

Round 2

Reviewer 2 Report

Dear authors,

if editors wanted narrative review it's up to them to make final decision. As reviewer I will follow the guidelines. As I stated it is is fine professional paper.  However, I am still not in favor that this is review! I saw that one reviewer suggested mini review. 

I am satisfy with result plagiarism analysis which is 21.8%.